# Dynamin phosphorylation controls optimization of endocytosis for brief action potential bursts

Moritz Armbruster[1,2], Mirko Messa[3,4], Shawn M Ferguson[3], Pietro De Camilli[3,4], Timothy A Ryan[1]*

[1]Department of Biochemistry, Weill Cornell Medical College, New York, United States; [2]The David Rockefeller Graduate Program, Rockefeller University, New York, United States; [3]Department of Cell Biology, Program in Neurodegeneration and Repair, Yale University School of Medicine, New Haven, United States; [4]Kavli Institute for Neuroscience, Howard Hughes Medical Institute, Yale University School of Medicine, New Haven, United States

**Abstract** Modulation of synaptic vesicle retrieval is considered to be potentially important in steady-state synaptic performance. Here we show that at physiological temperature endocytosis kinetics at hippocampal and cortical nerve terminals show a bi-phasic dependence on electrical activity. Endocytosis accelerates for the first 15–25 APs during bursts of action potential firing, after which it slows with increasing burst length creating an optimum stimulus for this kinetic parameter. We show that activity-dependent acceleration is only prominent at physiological temperature and that the mechanism of this modulation is based on the dephosphorylation of dynamin 1. Nerve terminals in which dynamin 1 and 3 have been replaced with dynamin 1 harboring dephospho- or phospho-mimetic mutations in the proline-rich domain eliminate the acceleration phase by either setting endocytosis at an accelerated state or a decelerated state, respectively.

*For correspondence: taryan@med.cornell.edu

**Competing interests:** The authors declare that no competing interests exist.

## Introduction

Synaptic transmission relies on a steady supply of release competent neurotransmitter-filled synaptic vesicles (SVs) to maintain information transmission in neural circuits. At small CNS nerve terminals the size of the recycling SV pool is often limited (*Harata et al., 2001*; *de Lange et al., 2003*; *Fernandez-Alfonso and Ryan, 2004*; *Kim and Ryan, 2010*; *Denker et al., 2011*) and therefore control of steps in SV recycling will likely prove important in setting overall synaptic performance. Given the importance of $Ca^{2+}$ in regulating exocytosis, the role of this ion in regulating SV recycling steps has been of significant interest for over 30 years (*Ceccarelli et al., 1979*; *Ceccarelli and Hurlbut, 1980*). At hippocampal and cortical nerve terminals synaptic vesicles are retrieved largely in a dynamin (*Ferguson et al., 2007*; *Raimondi et al., 2011*), clathrin (*Granseth et al., 2006*) and AP-2 dependent fashion (*Kim and Ryan, 2009*). Single vesicle retrieval studies (*Balaji and Ryan, 2007*) demonstrated that endocytosis occurs in a probabilistic fashion, and the mean endocytosis time constant for large bursts (100 AP 10 Hz) is a cell-wide property that can vary significantly from neuron to neuron (*Armbruster and Ryan, 2011*). Numerous studies have revealed a modulatory role for $Ca^{2+}$ in endocytosis at nerve terminals in different synaptic preparations (*Balaji et al., 2008*; *Sankaranarayanan and Ryan, 2001*; *von Gersdorff and Matthews, 1994*; *Wu et al., 2009*; *Yamashita et al., 2010*; *Yao et al., 2009*) although the precise steps at which $Ca^{2+}$ acts in this process have not been identified. The direction of the $Ca^{2+}$ modulation is also in dispute, some studies suggest an accelerating role of $Ca^{2+}$, (*Sankaranarayanan and Ryan, 2001*; *Wu et al., 2009*) while others

**eLife digest** Neurons communicate with each other at specialized junctions called synapses. When signals travelling along a neuron reach the presynaptic cell, this triggers small packages (vesicles) containing neurotransmitter molecules to release their contents into the synapse, and these molecules then cross the gap and bind to receptors on the postsynaptic neuron.

To release their cargo, individual vesicles fuse with the plasma membrane of the presynaptic neuron and form a 'pore' through which neurotransmitter molecules can leave the cell. However, to avoid running out of vesicles, the neuron must recycle and rebuild them through a process known as endocytosis. This involves recapturing the proteins that make up the synaptic vesicle and internalizing them back into the presynaptic terminal.

Exactly how endocytosis is regulated has been the subject of much debate in recent years. Now, Armbruster et al. have used fluorescent markers to study the timing of endocytosis in unprecedented detail. Observations of individual synapses reveal that when a series of action potentials (spikes of electrical activity) occurs in a neuron, endocytosis accelerates during the first few action potentials, and then slows. However, this acceleration was only detectable at a physiological temperature of 37°C—markedly higher than the 30°C at which synaptic endocytosis is typically studied.

The new study showed that acceleration of endocytosis depends on the phosphorylation status of dynamin, a mechano-chemical enzyme long known to be crucial for endocytosis, which helps to sever the connection between the endocytosing membrane and the surface of the cell. Phosphorylation is a common mechanism for controlling enzyme activity, and involves the addition of phosphate groups to specific amino acids by enzymes called kinases. Phosphatase enzymes reverse the process by removing the phosphate groups. Dynamin is usually phosphorylated at two specific amino acids, but when levels of calcium in the cell increase (as occurs during action potentials), a phosphatase called calcineurin dephosphorylates these sites. Using versions of dynamin that were either permanently phosphorylated or never phosphorylated, Armbruster et al. showed that a decrease in dynamin phosphorylation was required for the initial acceleration of endocytosis.

This type of regulation seems to optimize the recycling of vesicles to enable neurons to respond effectively to brief bursts of stimulation. Given that dynamin phosphorylation is conserved in evolution, it is likely that regulation of synaptic endocytosis is a key mechanism for ensuring the efficient functioning of the nervous system. Future research will investigate how calcium influx mediates the later slowing of endocytosis, and help to further unravel this previously unknown regulatory process.

suggest that $Ca^{2+}$ slows endocytosis (***Balaji et al., 2008***; ***von Gersdorff and Matthews, 1994***; ***Sun et al., 2002***; ***Sankaranarayanan and Ryan, 2000***). Although a number of proteins with $Ca^{2+}$ sensing domains have been implicated in SV endocytosis (synaptotagmin, calmodulin, calcineurin) direct links connecting endocytic behavior to consequences of $Ca^{2+}$ sensing have not been established. We took advantage of the ability to map endocytosis kinetics with high fidelity using a pHluorin-tagged synaptic vesicle protein to show that endocytosis has a pronounced acceleration phase with increasing number of action potentials used to elicit exocytosis. This initial acceleration phase is followed by a gradual slowing with increasing stimulus number. Our previous studies had missed this acceleration phase, in part due to the variability of endocytosis time constants across different neurons and in part owing to the fact that it is only readily apparent at physiological temperatures (37°C). We show that both phases of stimulus-dependent endocytosis are modulated by $Ca^{2+}$. The existence of two phases with opposite sign implies endocytosis has an optimal minimal value that is tuned by calcium-dependent processes.

Dynamin, a mechano-chemical enzyme that plays a key role in membrane fission (***Ferguson and De Camilli, 2012***), was identified in a search for proteins whose phosphorylation decreases upon $Ca^{2+}$ entry at nerve terminals (***Robinson and Dunkley, 1983***). Subsequent studies demonstrated two specific serines in dynamin's proline rich domain as substrates for the calcium-dependent phosphatase calcineurin and the proline-directed serine/threonine kinase CDK5 (***Graham et al., 2007***). In order to investigate the role of dynamin's phosphorylation in controlling endocytosis we made use of the fact

that eliminating the two major brain isoforms (dynamin 1 and 3) results in a dramatic (>10-fold) slowing in endocytosis kinetics (*Raimondi et al., 2011*). This strong phenotype allowed us to sensitively probe the ability of different dynamin isoforms to restore endocytic function in response to varying stimulus conditions. We show that mutating the two key phosphorylated serines in dynamin 1 to either alanine or aspartate both rescue the major endocytic defect of the dynamin 1/3 KO however they both eliminate the activity-dependent acceleration of endocytosis kinetics. These studies thus pinpoint dynamin 1 as a critical substrate in activity-dependent modulation of synaptic vesicle endocytosis and that it forms a basis for fine-tuning the retrieval process.

## Results

### Calcium-dependent slowing of endocytosis for stimuli >10 AP

We made use of pHluorin-tagged vesicular glutamate transporter (vG-pH) transfected into primary neurons to provide high sensitivity optical assays of synaptic vesicle endocytosis. vG-pH fluorescence is quenched by the acidic lumen of the synaptic vesicle (pH 5.6). Upon exocytosis the vesicular fluorescence increases ~20-fold and is requenched by reacidification after endocytosis (*Sankaranarayanan et al., 2000*). The endocytic time constant can be deconvolved from reacidification and measured from the fluorescence decay after stimulation (*Granseth et al., 2006*; *Balaji et al., 2008*) where it has been shown in numerous studies to follow simple single exponential decay kinetics over a broad range of stimulus conditions (*Balaji and Ryan, 2007*; *Armbruster and Ryan, 2011*; *Yao et al., 2011*; *Kwon and Chapman, 2012*; *Willox and Royle, 2012*). Changes in reacidification rates are only expected to minimally impact endocytosis time constant estimates (see 'Materials and methods') and for the relevant stimulus conditions asynchronous release would unlikely affect the measures of endocytosis (*Atluri and Ryan, 2006*; *Granseth et al., 2006*). We recently extended this technology to characterize endocytosis at individual boutons (*Armbruster and Ryan, 2011*) where the probe allows for many rounds of stimulation and recovery within a given field of view, thus allowing the same synaptic boutons to be probed many times with different stimulus conditions. Our earlier examination of stimulus dependence relied more heavily on analysis of ensemble behavior across many cells (*Balaji et al., 2008*) where we saw little evidence for modulation of endocytosis kinetics for stimuli <100 AP. Comparison of fluorescence recovery profiles for exocytosis triggered with a 100 AP and 10 AP at 10 Hz (at 30°C) at the same boutons however revealed that the exponential endocytic decay was slower for the larger stimulation (*Figure 1A*). Experiments carried out at an individual set of boutons for a range of stimuli (10, 25, 50, and 100 AP) showed that this trend was continuous, with gradual slowing of the endocytic time constant ($\tau_{endo}$) with increasing stimulus number (*Figure 1B*). The degree of slowing was most easily parameterized as a linear dependence on the number of action potentials used to drive exocytosis (*Figure 1B*). We examined this dependence on stimulus number across collections of boutons from many individual neurons (N = 44) and found both the degree of slowing (i.e., the slope in s/AP) and the extrapolated fastest time constant expected at 1 AP (the intercept) to vary significantly across neurons (*Figure 1C*) but were uncorrelated with each other. The mean degree of slowing for the population of neurons was 0.058 ± 0.004 s/AP.

Comparisons of the slowing behavior in the same synapses for stimuli delivered in 4 mM compared to 2 mM external $Ca^{2+}$ demonstrated that this inhibition of endocytosis appears to be enhanced under conditions that would lead to more elevated intracellular calcium (*Figure 1D*). These experiments revealed that the activity-dependent slowing was always steeper when stimuli were delivered in the higher $Ca^{2+}$ concentration. On average (n = 9) the slope increased by ~700% (*Figure 1E*) while the intercept was unchanged. The difference in slopes is not dependent simply on examining how $\tau_{endo}$ changes with stimulus number and can be readily observed if one alternatively examines the relationship of $\tau_{endo}$ vs exocytosis (*Figure 2—figure supplement 1*). We further examined a possible influence of $Ca^{2+}$ on the activity-dependence of endocytosis by measuring the activity dependence before and after loading nerve terminals with the $Ca^{2+}$ chelator EGTA-AM which at 30°C (the temperature used for these experiments) still allows for significant exocytosis. These experiments revealed that buffering intracellular $Ca^{2+}$ slowed endocytosis compared to control across stimuli but eliminated slowing as a function of stimulation. Additionally incubation with EGTA-AM unmasked a modest acceleration phase for the lowest stimuli (*Figure 1F*). Similar to the impact of varying external $Ca^{2+}$, the impact of EGTA on stimulus-dependent slowing was independent of any changes in exocytosis (data not shown).

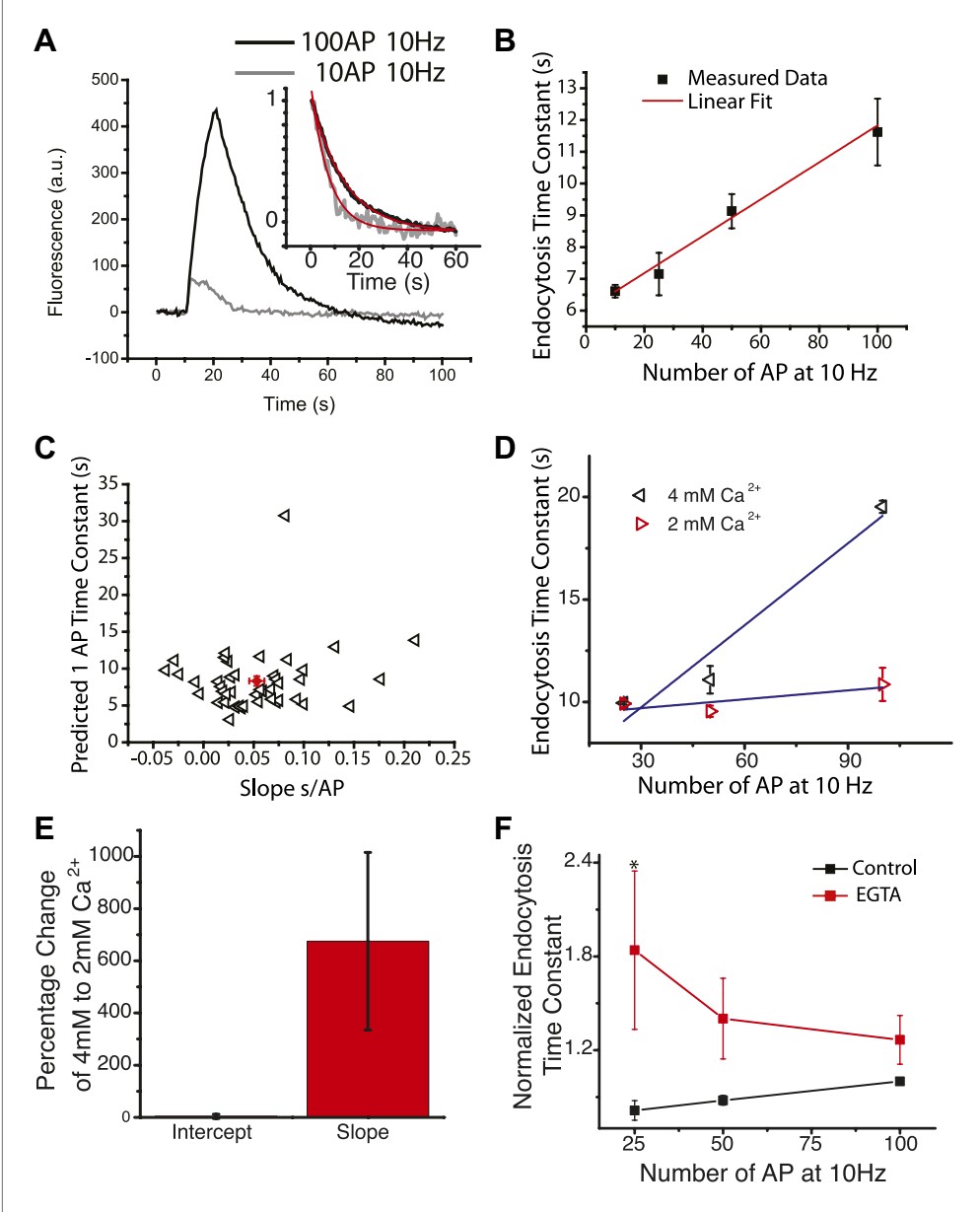

**Figure 1**. Calcium slows endocytosis at 30°C. (**A**) Endocytosis decays from a 100 AP 10 Hz run (black) and 10 AP 10 Hz runs (2–3 runs averaged) (gray), inset shows endocytosis phase fit with a single exponential decay (red) to measure endocytosis time constant = 13.9 ± 0.10 s, 6.8 ± 0.3 s 47 boutons. adj. R-square of fits 0.95, 0.997 respectively. (**B**) An example cell probed multiple times at 10, 25, 50, 100 AP at 10 Hz at 30°C, fit with a linear dependence with a slope of 0.058 ± 0.004 s, a predicted 1 AP time constant of 6.09 ± 0.11 s. (**C**) Across 44 cells, the slope (s/AP) is plotted against the predicted 1 AP time constant (s). Average slope 0.053 ± 0.008 s, average predicted 1 AP time constant 8.31 ± 0.64 s. There is a CV of 100% in the slope and 52% in the predicted 1 AP time constant. (**D**) Endocytosis for 25, 50 and 100 AP delivered in 2 mM and 4 mM external $Ca^{2+}$ for the same set of boutons from one cell. Each condition was probed 1–4 times and averaged over 46 ROIs. Slope of 2 mM is 0.01 ± 0.01 s/AP, 4 mM is 0.13 ± 0.03 s/AP. The intercept of 2 mM is 9.3 ± 0.7 s, 4 mM is 5.7 ± 2.0 s. (**E**) Across 10 cells the percentage change in slope and predicted 1 AP time constant when changing from 2 mM to 4 mM external $Ca^{2+}$ slope and predicted 1 AP time constant. The change in slope is significantly different from 0, (log corrected one sample t-test p<0.004). (**F**) 25, 50, and 100 AP at 10 Hz endocytosis time constant compared before and after 90 s load of 100 μM EGTA-AM reveals an acceleration phase of endocytosis for low stimulus number (25 AP significantly different before and after EGTA treatment, paired sample *t*-test p<0.03, N = 6 cells).

## Endocytosis optimized for brief bursts of AP at physiological temperature

The continuous slowing of $\tau_{endo}$ with increasing stimulus number in the 10–100 AP range predicts that the value of $\tau_{endo}$ for single AP stimulation (i.e., the predicted intercept in *Figure 1B*) would be the fastest of any stimulus. However we and others have previously shown that the endocytic recovery for single AP stimuli were similar to that obtained for much larger stimuli (*Granseth et al., 2006*; *Balaji and Ryan, 2007*). Additionally our experiments with EGTA buffering suggests that endocytosis would have the opposite behavior in the low stimulus regime, given that smaller stimuli lead to less total $Ca^{2+}$ entry. To directly examine this we systematically examined endocytosis following single AP stimulation as well as following 25, 50 and 100 AP in the same boutons. (*Figure 2A,B*). The observed single AP $\tau_{endo}$ was slower than for that obtained at 25 AP (*Figure 2A*), as well as than that predicted from the linear regression at higher stimulus levels (*Figure 2B,C*). Taken together these data thus demonstrate two distinct phases of activity-dependence: an acceleration of endocytosis for stimuli >1 AP and a slowing for stimuli > approximately 10 AP. This predicted acceleration phase agrees with the results of our EGTA experiments (*Figure 1F*) where buffering intracellular $Ca^{2+}$ during bursts of AP revealed an acceleration phase for lower stimulus numbers.

The existence of these two opposite phases of $Ca^{2+}$ modulation on endocytosis thus appears critical in determining the endocytosis time constant and would likely be influenced by any factors that impact the degree of calcium entry and accumulation during AP bursts. One such critical physiological parameter is temperature. The experiments above were all carried out at 30°C. Although this temperature was chosen to provide a practical signal to noise ratio for experiments, given the known temperature-dependence on $Ca^{2+}$ handling (*Sabatini and Regehr, 1998*) and action potential waveforms (*Hlavova et al., 1970*) we reasoned that endocytic optima might be hard to pinpoint unless one worked at physiological temperature. For this reason we explored the activity-dependent behavior of endocytosis for a range of stimuli (between 1 AP and 100 AP at 10 Hz) at 37°C and 30°C (*Figure 2D*).

Given the large variability from cell to cell in endocytosis time constant, in order to compare stimulus-dependent variation across cells in each experiment the data were internally normalized to the value for $\tau_{endo}$ obtained for 100 AP. Data normalized in this fashion revealed the expected slowing behavior between 10 AP and 100 AP. Although the acceleration between 1 AP and 10 AP was readily apparent for individual cells (*Figure 2B*) at 30°C, on average this acceleration of endocytosis was not easily resolvable at this cooler temperature (*Figure 2D*). However these experiments revealed a clear and robust acceleration phase for endocytosis at physiological temperature with stimuli ranging from 5 AP to 25 AP and milder slowing phase for stimuli above this than that seen at 30°C, with a minimum endocytosis time constant in the vicinity of 25 AP at 10 Hz. At physiological temperature, on average, endocytosis accelerated by 42 ± 7% (N = 9) going from 5 AP to 25 AP and then slowed by 18 ± 4% (N = 8) at the 100 AP stimulus level. The prominent slowing phase of endocytosis present at 30°C however was still observed at 37°C but it required larger stimuli (300 AP) to show obvious (55 ± 20% [N = 7]) slowing when compared to 100 AP 10 Hz. The more pronounced acceleration of endocytosis in the low stimulus regime implicates calcium as a likely modulator for mediating the acceleration as intracellular calcium would likely accumulate during brief bursts. We tested this notion explicitly by comparing the endocytic time constant for single AP stimulation at 2 mM and 4 mM $Ca^{2+}$ at 37°C. These experiments showed that increasing $Ca^{2+}$ accelerated the post-stimulus endocytosis kinetics significantly (*Figure 2E*). The acceleration is more clearly present at physiological temperature, we expect through reduced $Ca^{2+}$ influx on a per action potential basis. Conversely at a higher stimulation (30 Hz) frequency the acceleration occurs over a very small stimulus range (*Figure 2—figure supplement 2*). This makes the optimum stimulus for endocytosis hard to discern, but shows a larger relative impact on endocytosis relative to single AP responses (*Figure 2F*).

## Acceleration of endocytosis has a persistence time of 20–30 s

Increasing the stimulation from 5 AP to 10 AP at 10 Hz accelerates the endocytosis time constant (*Figure 2D*), which cannot be explained by changes in the reacidifcation kinetics (see 'Materials and methods'). However it is presumably a $Ca^{2+}$ dependent process (*Figure 1F*). In order to narrow down possible mechanisms of $Ca^{2+}$ action in this process we sought to determine how long the effect of a burst of stimulation would last in accelerating endocytosis. To examine this we designed a protocol where we examined endocytosis following a 5 AP burst delivered at different inter-burst intervals for five total bursts. For an inter-burst interval of 0 s it is the equivalent to looking at a single prolonged burst of 5, 10, 15, or 25 AP at 10 Hz. A representative example using a 30 s inter-burst interval is shown

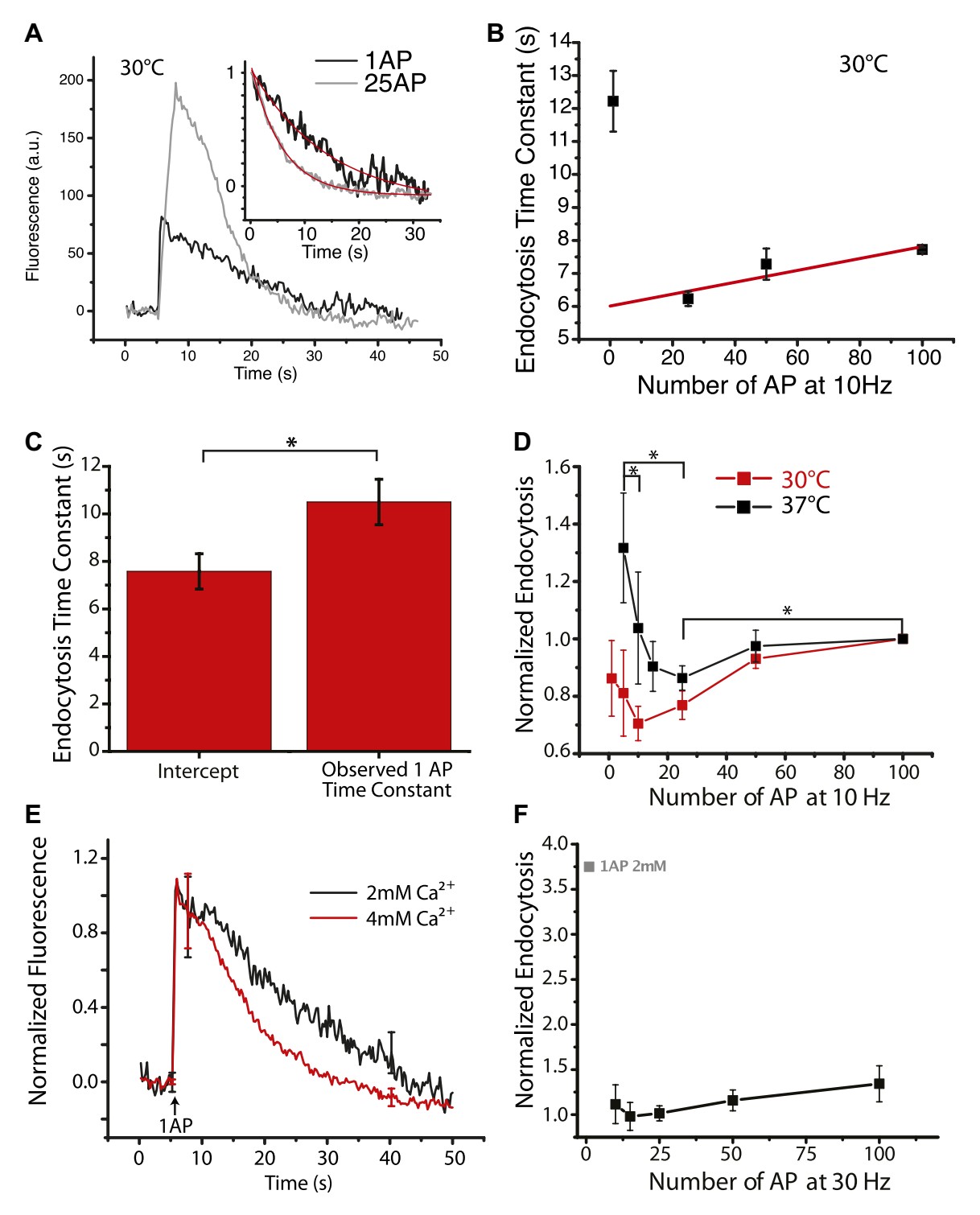

**Figure 2**. Acceleration of endocytosis for small stimuli. (**A**) Individual example traces of a cell probed with 1 AP (black) and 25 AP at 10 Hz (gray), inset shows endocytosis phase with their fits (red) at 30°C. 1 AP time constant $\tau$ = 12.36 ± 1.1 s, 25 AP 10 Hz time constant $\tau$ = 8.57 ± 0.67 s, adj. R-square of fits 0.93, 0.99 respectively. (**B**) The same example cell probed at 1, 25, 50, 100 AP at 10 Hz at 30°C, with a linear fit to the 25, 50, 100 AP data. Each point is an average of 2–3 runs based on 76 ROIs. Predicted 1 AP time constant = 6.55 ± 0.54 s. (**C**) Across 10 cells, the predicted 1 AP time constant based upon linear fit to 25, 50, 100 AP 10 Hz data compared to observed 1 AP time constant. The difference is significant paired sample *t*-test p<0.01. (**D**) At physiological temperature, (37°C black) probing 5, 10, 15, 25, 50, 100 AP at 10 Hz, normalized for 100 AP 10 Hz tau for each

*Figure 2. Continued on next page*

*Figure 2. Continued*

cell, showing the acceleration of endocytosis. N = 8 cells. 5 AP (maximum) vs 25 AP (minimum) is significant p<0.01, 25 AP vs 100 AP is significant p<0.03 paired sample *t*-tests, 5 AP vs 10 AP is significant p<0.009 paired sample *t*-tests. At 30°C (red) there is no significant acceleration over a similar range of stimuli 1 AP vs 10 AP (minimum) p>0.05 paired *t*-test. N = 8, 7, 12, 14, 12, 14 cells for 1, 5, 10, 25, 50, and 100 AP 10 Hz respectively. (**E**) Paired comparisons of 1 AP stimulation at 2 mM and 4 mM extracellular $Ca^{2+}$ showing a $Ca^{2+}$ dependent acceleration (N = 6 cells, significant difference in remaining fluorescence at 20 s paired *t*-test p<0.05). (**F**) Acceleration of endocytosis measured for 30 Hz AP bursts normalized to the value obtained at 15 AP, but including the relative value measured for 1 AP (2 mM) in **D**. N = 3, 4, 5, 4, 4 cells for stimuli 100, 50, 25, 15 10 AP at 30 Hz.

The following figure supplements are available for figure 2:

**Figure supplement 1**. Comparing the effects of 2 mM and 4 mM external $Ca^{2+}$ on the slopes of endocytosis corrected for changes in exocytosis.

**Figure supplement 2**. Mapping the acceleration notch curve at 30 Hz, at 100, 50, 25, 15, 10 AP.

in *Figure 3A*. These experiments revealed that the acceleration of endocytosis caused by 5 AP persists for at least 15 s, but is lost for intervals >30 s. To compare results across many cells data for each cell were normalized to the value of $\tau_{endo}$ for a single 5 AP burst for each experiment and inter-burst intervals of 0 s (*Figure 3C*), 15 s (*Figure 3D*), 20 s (*Figure 3E*) and 30 s (*Figure 3F*) were examined for many cells. For the 15 s inter-burst interval (*Figure 3D*) we relied on characterizing the endocytic time scale as (1/rate) as the time frame was too compressed to allow accurate exponential fitting. These experiments showed that endocytosis was accelerated for a continuous 10 AP burst compared to a 5 AP (*Figure 3C*, similar to *Figure 2D*). The impact of a single 5 AP burst on accelerating endocytosis however persisted for at least 20 s, as $\tau_{endo}$ for a second 5 AP burst was accelerated to the same extent as providing a continuous 10 AP burst even if the second burst was delivered 20 s later. Measurements for a 30 s inter-burst interval however showed no significant acceleration. Example traces of endocytosis for the first 2 bursts for 20 s and 30 s intervals from two different cells (*Figure 3B*) illustrate this point. These experiments show that the acceleration persists for ~20 s between stimuli but is dissipated after ~30 s. Given that elevations in intracellular $Ca^{2+}$ following a 5 AP burst decays on much faster time scales (<1 s) these data imply that $Ca^{2+}$ is likely acting through a second messenger system to control endocytosis in the acceleration phase.

## Dynamin 1 dephosphorylation at serines 774/778 mediate activity-dependent acceleration of endocytosis

A potential candidate for an endocytic mechanism that is mediated by a second messenger downstream of $Ca^{2+}$ is the control of the phosphorylation state of dephosphin proteins. Dynamin 1, the founding member of the dephosphin family and by far the most abundant neuronal dynamin, is constitutively phosphorylated at serines 774 and 778 by Cdk5 and dephosphorylated in a stimulus- induced manner at the same sites by the $Ca^{2+}$ dependent phosphatase calcineurin (*Liu et al., 1994*). A major effect of this phospho-regulation is to control the interaction of dynamin 1 with syndapin, whose binding to dephospho-dynamin is abolished by phosphorylation (*Anggono et al., 2006*; *Anggono and Robinson, 2007*). Interestingly, dephosphorylation of dynamin after a stimulatory burst has been shown to persist for ~40 s under certain stimulus conditions (*Robinson et al., 1994*). We previously showed that dynamin 1/3 DKO mice have severely impaired endocytosis; however, nerve terminals lacking these major dynamin isoforms are still able to undergo multiple rounds of vesicle recycling (*Raimondi et al., 2011*). We re-examined endocytosis in dynamin 1/3 DKO neurons at 37°C which showed a very similar phenotype to our previous studies (*Figure 4A*): the absence of both dynamin 1 and 3 results in a severe impairment of endocytosis, which can be fully rescued by re-introduction of dynamin 1. These studies in mouse neurons necessitated using cortical neurons rather than hippocampal neurons due to the small size of the hippocampus in newborn mice, and the smaller total neuronal population. We therefore reexamined the stimulus dependence of endocytosis in mouse cortical neurons. These experiments showed very similar behavior as rat hippocampal neurons with the exception that the slowing phase was much less pronounced and was only evident at much higher stimulus levels (see below). The acceleration phase however was readily apparent in both the mouse cortical neuron controls and dynamin 1/3 DKO mouse cortical neurons in which dynamin 1 has been reintroduced (dynamin 1 rescue, *Figure 4B*) showing a 31 ± 12% (N = 9) and 21 ± 8% (N = 7)

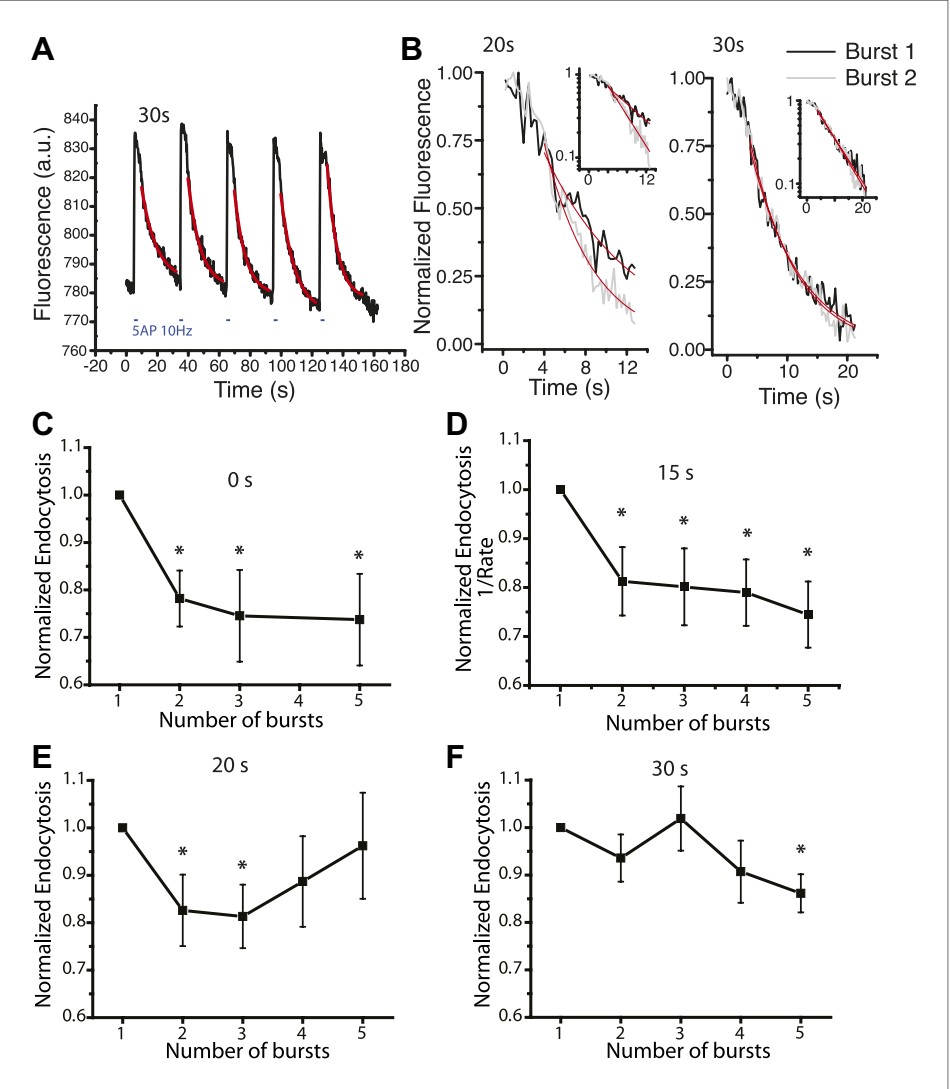

**Figure 3**. Persistence time of endocytic acceleration. (**A**) A sequence of 5 bursts of 5 AP 10 Hz with inter-burst interval of 30 s apart from one cell averaged over 4 runs, each decay is fit with an exponential decay. Endocytosis time constants = 9.7 ± 0.8 s, 7.9 ± 0.5 s, 7.2 ± 0.4 s, 9.2 ± 0.5 s, 6.7 ± 0.3 s respectively for pulses 1–5. (**B**) Example endocytic decays from pulses 1 and 2 with 20 s and 30 s spacing illustrating the acceleration of endocytosis for 20 s inter-burst interval spacing: 20 s time constants 8.6 ± 0.5 s and 4.8 ± 0.2 s for first and second burst respectively, adj. R-square of fits 0.88, 0.93 respectively; 30 s time constants 8.3 ± 0.2 s and 7.7 ± 0.2 s first and second burst respectively, adj. R-square of fits 0.96 and 0.97 respectively. Data were averaged over 5–12 runs across 30–50 boutons. (**C**) Inter-burst interval of 0 s, based upon data from **Figure 2D**, normalized to 5 AP 10 Hz shows significant acceleration of endocytosis, p<0.02 for 10 AP, 15 AP, and 25 AP 10 Hz pulses n = 8 cells. (**D**) 15 s spacing with linear fits for the decays, plotting 1/Rate of endocytosis, showing significant acceleration of endocytosis compared to the first burst for all subsequent bursts, p<0.03 for bursts 2–5. n = 11 cells. (**E**) 20 s spacing, fit with exponential time constants, bursts 2 and 3 are significantly accelerated compared to the first burst p<0.05, n = 9 cells. (**F**) 30 s inter-burst interval with exponential fits, only the fifth burst is significantly accelerated compared to the first burst p<0.04, n = 9 cells. All tests one sample *t*-test.

acceleration between 10 AP and 100 AP in the dynamin 1 rescue and control respectively (compared to 42 ± 7% [N = 9]) for rat hippocampal neurons. Consistent with the need to use greater stimulation to see the slowing phase, mouse cortical neurons appear to have their acceleration phase shifted slightly to higher stimulus numbers compared to rat hippocampal neurons as well. In order to examine the possible role of dynamin 1 dephosphorylation in mediating acceleration of endocytosis we mutated

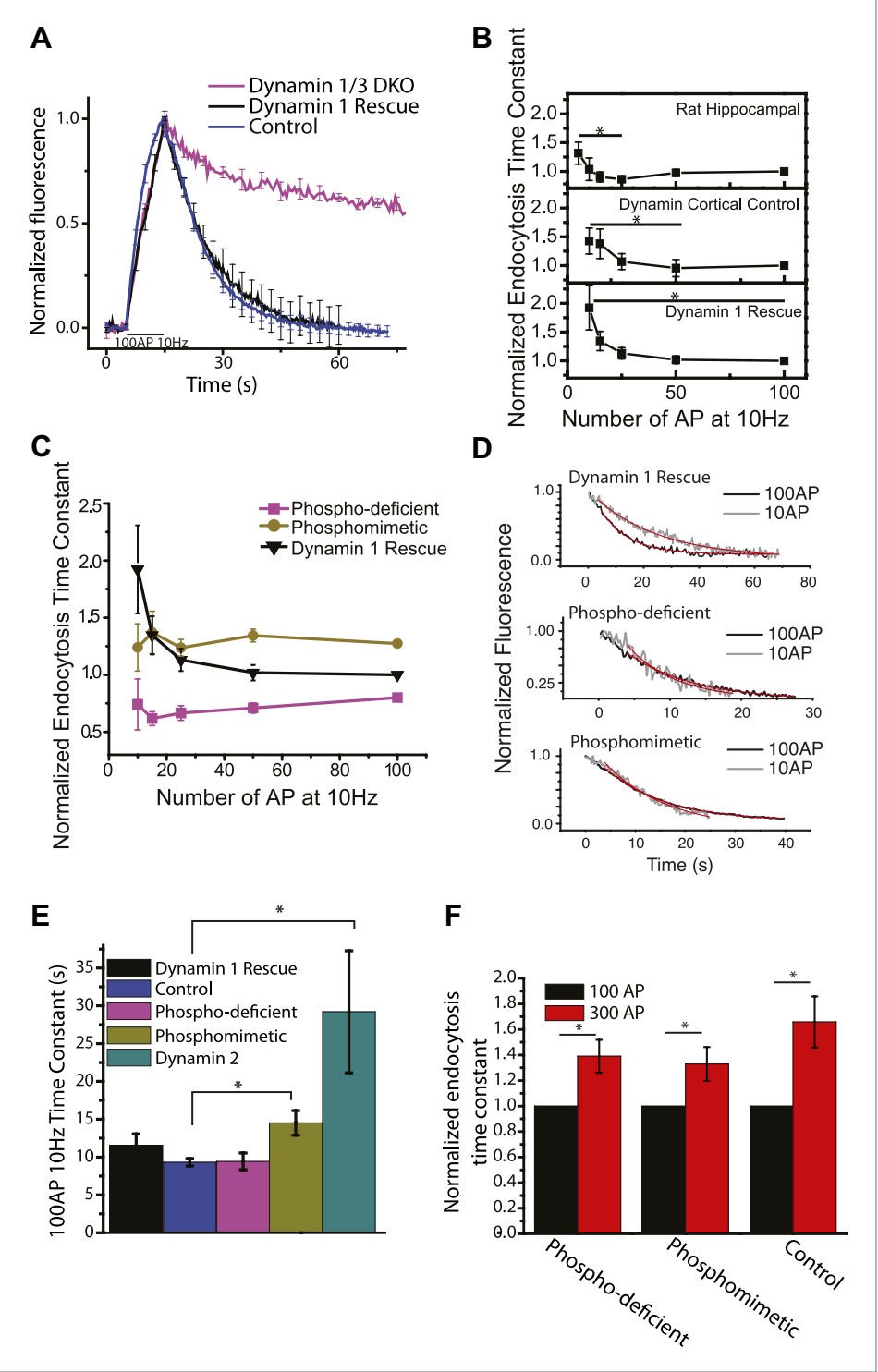

**Figure 4**. Dephosphin control of endocytic acceleration. (**A**) 100 AP 10 Hz stimulation of dynamin 1/3 DKO, dynamin 1 rescue expressed in DKO, or the dynamin 1 Het dynamin 3 KO control genotype at 37°C. n = 5, 7, 10 cells respectively. (**B**) Comparison of endocytosis acceleration in rat neurons (top) cortical mouse neurons (middle) and cortical mouse dynamin 1/3 DKO neurons rescued with dynamin 1. For each cell data is normalized to the value obtained for 100 AP 10 Hz, N = 8 rat hippocampus, 7 mouse cortical, 9 dynamin 1 rescue. Mouse cortical 10 AP compared to 50 AP (minimum) is significant p<0.03; dynamin 1 rescue 10 AP compared to 100 AP (minimum) is significant p<0.05 paired sample *t*-tests. (**C**) Endocytosis vs stimulation for dynamin 1/3 DKO rescued with the full
*Figure 4. Continued on next page*

*Figure 4. Continued*

length dynamin 1 (black, replotted from **B**, bottom), phospho-deficient, S774/8A (pink), or the phosphomimetic S774/8D mutants of dynamin 1 (gold) shows that mutations at these serines block acceleration and lock endocytosis in a fast or slow state. Individual traces are normalized to the 100 AP 10 Hz value for the dynamin 1 rescue. N = 9 cells dynamin 1 rescue, N = 6 cells phospho-deficient, N = 8 cells phosphomimetic. (**D**) Example traces of 10 AP compared to 100 AP for dynamin 1 rescue (adj. R-square of fits 0.97, 0.97 100 AP, 10 AP respectively), phospho-deficient rescue (adj. R-square of fits 0.98, 0.89 100 AP, 10 AP respectively), and phosphomimetic rescue (adj. R-square of fits 0.999, 0.97 100 AP 10 AP respectively) of dynamin 1/3DKO showing the lack of acceleration for the phosphorylation mutants. Single cells based upon 30–50 boutons and 1–3 runs. Time constants of decays: dynamin 1 rescue 25.1 ± 0.4 s and 9.7 ± 0.4 s, phospho-deficient rescue 6.8 ± 1.2 s and 9.4 ± 0.5 , phosphomimetic rescue 12.4 ± 1.1 s and 12.0 ± 0.1 s for 10 AP and 100 AP respectively. (**E**) The 100 AP 10 Hz time constants of the dynamin 1/3 DKO rescued with dynamin 1, phosphorylation mutants and dynamin 2. Phosphomimetic is significantly different from control KS-test p<0.003. Dynamin 2 is significantly different from control KS-test p<0.0004. N = 7, 10, 9, 12, and 11 cells respectively. (**F**) Paired 100 AP and 300 AP 10 Hz stimulus, endocytosis time constants for the phosphomutants and control. Data normalized to 100 AP 10 Hz paired *t*-test p<0.03 for all conditions. 300 AP $\tau_{endo}$ phospho-deficient 1.39 ± 0.13 s, phosphomimetic 1.33 ± 0.13 s, control 1.66 ± 0.20 s. N = 9, 15, 9 cells respectively.

The following figure supplements are available for figure 4:

**Figure supplement 1**. Exocytosis and pool size controls for dynamin rescues.

serines 774 and 778 to either alanine or aspartate, used these isoforms of dynamin 1 to rescue the endocytic defect in dynamin 1/3 DKO neurons and examined the stimulus-dependence of endocytosis. Previous studies have shown that mutations of these serines to alanine and aspartate mimic the dephosphorylated and phosphorylated states respectively with respect to dynamin 1's ability to bind syndapin (*Anggono et al., 2006*). Although both phosphomimetic (S774/778D) and phospho-deficient (S774/778A) mutants could efficiently rescue the severe dynamin 1/3 DKO endocytosis defect, neither showed any activity dependent acceleration (*Figure 4C,D*). The phosphomimetic form was effectively locked in a slower endocytic state, while the phospho-deficient form was faster across all stimuli (*Figure 4C,E*). Thus, neurons expressing mutants of dynamin 1 that lock the phosphorylation sites in a specific state do not undergo stimulus-dependent regulation and constitutively proceed at overall faster (phospho-deficient mutant) or slower (phospho-mimetic mutant) endocytic speeds. Dynamin 2, which has different activity dependent phosphorylation sites (*Chircop et al., 2011*), was only able to partially rescue the 100 AP 10 Hz time constant and was not tested further. Increasing the stimulus number from 100 AP to 300 AP showed a slowing of endocytosis for all conditions similar to the relationship described at 30°C (*Figure 4F*). Using the bafilomycin method (*Sankaranarayanan and Ryan, 2001*) we detected no difference in the size of the recycling pool of synaptic vesicles or in the rate of exocytosis between control, dynamin 1, phosphomimetic, or phospho-deficient rescue (*Figure 4—figure supplement 1*).

## Discussion

Using high-sensitivity pHluorin assays of synaptic vesicle endocytosis for individual neurons we revealed the presence of two phases of stimulus and $Ca^{2+}$ dependence of synaptic vesicle endocytosis: an acceleration phase prominent for small stimuli and a slowing phase prominent for larger stimuli. Our experiments made use of pHluorin-tagged vGlut expressed in dissociated neurons. Although it is likely that these data sets arise from a mixture of gabaergic and glutamatergic neurons, we previously showed that these two neuron classes differ little in endocytic behavior, even when vGlut is expressed in a gabaergic neuron (*Armbruster and Ryan, 2011*), perhaps owing to the tracer-level expression of this probe (*Balaji and Ryan, 2007*). While the vGlut-pHluorin reporter only tracks the internalization of vGlut, previous synaptic vesicle endocytosis studies have shown SynaptopHluorin (Vamp2), Synaptophysin-pHluorin, and Synaptotagmin-pHluorin all showing the same kinetics as vGlut-pHluorin as well as the same dependence on the clathrin adaptor AP-2 (*Kim and Ryan, 2009*). Our data helps reconcile numerous reports on the $Ca^{2+}$ sensitivity of synaptic vesicle endocytosis. $Ca^{2+}$ has long been implicated in controlling synaptic vesicle endocytosis and numerous calcium-sensing proteins have been implicated in synaptic vesicle retrieval including calcineurin, synaptotagmin, and calmodulin (*Poskanzer et al., 2003*; *Nicholson-Tomishima and Ryan, 2004*; *Poskanzer et al., 2006*; *Yao et al., 2011*; *Yao and Sakaba, 2012*). While inhibiting or mutating these $Ca^{2+}$ sensors impact synaptic vesicle

endocytosis, to our knowledge these putative modulators of endocytosis had not been directly linked to demonstrated modulation of endocytosis kinetics. Our experiments show that dynamin 1 phosphorylation sites that were previously demonstrated to be calcineurin substrates are critical specifically for the acceleration phase during brief action potential bursts. The dephosphorylation accelerates the kinetics of endocytosis without fundamentally changing the endocytic properties suggesting that this represents a tuning of the mechanism rather than a distinct pathway. The dynamin 1/3 DKO provides the ideal background to test the significance of the phosphorylation sites, as it is necessary to also remove dynamin 3 since it has a similar phospho-box motif as dynamin 1 (*Larsen et al., 2004*). Although recent studies implicate dynamin's phosphorylation in controlling bulk endocytosis, analysis in dynamin 1/3 DKO nerve terminals indicates that this form of endocytosis proceeds unimpaired in the absence of dynamin (Yumei Wu, Shawn Ferguson and Pietro De Camilli, in preparation).

Analysis of single synaptic vesicle retrieval demonstrated that endocytosis is stochastic and the mean time for endocytosis is determined by a single rate-limiting step (*Balaji and Ryan, 2007*). As dynamin is considered to be a critical enzyme in endocytic membrane fission (*Ferguson and De Camilli, 2012*), the fact that endocytosis kinetics can be accelerated by dephosphorylation of dynamin 1 suggests that under these conditions membrane fission, or another dynamin-dependent event, is the rate limiting step in endocytosis. Phosphorylation of dynamin at serines 774 and 778 abolishes the interaction with the F-BAR protein syndapin and dephosphorylation promotes it (*Anggono et al., 2006*; *Anggono and Robinson, 2007*). Our data therefore suggest that the cooperative participation of syndapin and dephosphorylated dynamin improves the efficiency of the endocytic process: when dynamin cannot bind syndapin (phosphomimetic mutation) no acceleration occurs, while when dynamin can interact with syndapin even at rest (phospho-deficient mutation), endocytosis is already fast and no stimulus-dependent acceleration occurs. Only once further atomic-detail of the basis of syndapin's interaction with dynamin have been revealed will it be possible to test specifically the potential role of syndapin in this acceleration. Consistent with this scenario however, genetic ablation of syndapin 1, the major syndapin isoform expressed in brain, revealed a number of pleotropic phenotypes consistent with a failure to properly recruit dynamin to membranes (*Koch et al., 2011*) and syndapin is one of the proteins whose levels are more strongly reduced in mice that lack dynamin 1 and 3 (*Raimondi et al., 2011*).

The $Ca^{2+}$ dependent slowing phase we observed was more prominent at lower temperatures and agrees well with previous findings that show $Ca^{2+}$ inhibition of endocytosis (*Balaji et al., 2008*; *von Gersdorff and Matthews, 1994*; *Sun et al., 2002*). The $Ca^{2+}$ sensor for this mechanism is not known, although we showed that it is not affected by setting dynamin's phosphorylation state at the calcineurin sites. Previously a number of studies examined the role of the specific enzymes, calcineurin and CDK5, that control dynamin phosphorylation (*Liu et al., 1994*; *Tan et al., 2003*; *Tomizawa et al., 2003*), in controlling vesicle recycling. Although manipulation of these enzymatic activities impacted synaptic vesicle endocytosis, it is seems likely that these manipulations do not solely alter dynamin activity given that they target numerous substrates. Recent studies have recently shown for example that both these enzymes profoundly modulate $Ca^{2+}$ influx (*Kim and Ryan, 2013*) precluding simple interpretations of such pharmacological manipulations on endocytosis.

Our data indicate that at 37°C synaptic vesicle retrieval appears to be optimized for brief action potential bursts and that the basis for this tuning is based on a balance of calcineurin activation vs an additional calcium-dependent inhibitory effect that dominates during prolonged stimulation. Furthermore dynamin appears to be centrally important in this short term optimization as preventing changes in the phosphorylation at two key serines in dynamin 1 eliminates this tuning of endocytosis. This data compels one to speculate that cells might tune their endocytic profile to be optimized for individual firing patterns. The tuning could be achieved by altering the balance of CDK5 and calcineurin activity as has been demonstrated for certain forms of homeostatic plasticity (*Kim and Ryan, 2010*), or by altering routes of $Ca^{2+}$ entry or clearance. These phosphorylation sites in dynamin are well conserved across species with a nervous system suggesting that that endocytic optimization is a fundamental property of nervous system function.

## Materials and methods

### Cell culture and imaging

Hippocampal CA3–CA1 regions were dissected from 1- to 3-day-old Sprague Dawley rats, dissociated, and plated onto poly-ornithine-coated glass and grown for 14–26 days as described previously

(*Ryan, 1999*). For experiments utilizing the dynamin knockout mice and littermate controls (dynamin 1 het, dynamin 3 KO) cortexes were dissected from postnatal 0 to 1-day-old mice were dissociated and plated onto poly-ornithine coated glass as previously described (*Ferguson et al., 2007*; *Raimondi et al., 2011*). Cultures were transfected with calcium-phosphate 7–8 days after plating and imaging was performed 13–26 days after plating (5–18 days after transfection). The reporter used was a chimera of the pH sensitive GFP, pHluorin and the vesicular glutamate transporter made by the Voglmaier lab (UCSF). Constructs for human dynamin 1 (aa spliceform), rat dynamin 2-mRFP (AAB spliceform), human dynamin 1 S774/778A (aa spliceform), and human dynamin 1 S774/778D (aa spliceform) were used. Between species (mouse/rat) and (mouse/human) there is >99% amino acid identity.

Coverslips were mounted in a rapid-switching, laminar-flow perfusion and stimulation chamber (volume ~75 µl) on the stage of a custom-built laser-illuminated epifluorescence microscope. Cells were perfused with a solution containing in mM: 119 NaCl, 2.5 KCl, $2CaCl_2$, $2MgCl_2$, 25 HEPES (buffered to pH 7.4), 30 glucose supplemented with 10 µM 6-cyano-7-nitroquinoxaline-2,3-dione (CNQX), and 50 µM D,L-2-amino-5-phosphonovaleric acid (AP5). For experiments involving 4 mM $Ca^{2+}$ tyrodes solution, $CaCl_2$ was swapped for $MgCl_2$. All chemicals were obtained from Sigma-Aldrich (St Louis, MO). Due to the low surface fraction of vG-pH (*Balaji and Ryan, 2007*), we gave brief bursts with 6 APs at 30 Hz every 4 s to find transfected cells in a dish. Identity of genotype and transfected plasmids were known to the investigator when performing imaging. Perfusion was kept between 75–250 µl per minute to ensure prolonged cell survival. Cells were imaged either at 30°C, or 36.8°C by heating the microscope objective with a flexible resistive heater (Omega, Stamford, CT) utilizing an on–off controller (Minco, Minneapolis, MN), which maintained the temperature at the objective within ±0.1°C as readout by a 100 Ω platinum thermistor (Minco). Cells were illuminated utilizing a 488 nm diode pumped solid state laser (Coherent, Santa Clara, CA), shuttered using an acousto-optic modulation during all periods without data acquisition. Fluorescence excitation and collection was through a 40X 1.3 NA Fluar Zeiss objective using 515–560 nm emission and 510 nm dichroic filters (Chroma, Bellows Falls, VT) and a 1.6X Optivar tube lens. Laser power at the back aperture was ~1 mW, imaging onto a Andor iXon+ (model number DU-897E-BV) back-illuminated electron-multiplying charge coupled device camera. Action potentials were evoked by passing 1 ms current pulses, yielding fields of ~10 V/cm via platinum-iridium electrodes from an Isolated current stimulator (World Precision Instruments, Sarasota, FL).

## Image and data analysis

Images were analyzed in ImageJ (http://rsb.info.nih.gov/ij/) using a custom-written plugin (http://rsb.info.nih.gov/ij/plugins/time-series.html). 2 µm diameter circular ROIs were placed on all varicosities based upon the ΔF image of a 100 AP 10 Hz run, between 25–120 ROIs were used per cell. Only boutons that did not split or merge, remained in focus and responded throughout all trials were chosen. All small stimuli were additionally averaged over several rounds (up to 10) of stimulation to increase the signal to noise before fitting. All fitting was done using OriginPro (OriginLab, Northampton, MA) with the Levenberg-Marquardt algorithm. Fits of the endocytosis time constant were single exponential decays with a temporal offset for reacidification of ~2–3 s at 36.8°C and ~5 s at 30°C as described previously (*Balaji and Ryan, 2007*). To assess the contribution of changing reacidification times to our endocytosis fits, we ran simulations based upon the biexponential model of synaptic vesicle endocytosis (*Granseth et al., 2006*) and our fitting protocols. A change in reacidification from 1.6 s to 2.6 s introduces a +8% error in the measurement of endocytosis for a 10 s time constant, suggesting that it is unlikely to explain our observations. In general, fits were conducted on the ensemble average of each run, or multiple runs for very small stimuli. All fits were visually inspected and we did not observe deviation from the single exponential characteristic. For 1 AP data at physiological temperature the remaining fluorescence 15 s after the end of stimulation is quantified as a more robust measure given the low signal to noise and very slow decays of some traces. For 30 s and 20 s spacing between bursts (*Figure 3*) decays were fit with single exponential decays with constant time windows for each burst. For 15 s spacing, there is insufficient time to carry out a robust fit to a single exponential. For these experiments endocytic performance was estimated by a linear fit to the decay (Rate) and plotted as 1/Rate. Time constants or 1/Rate measures were normalized for the measure of the first burst in the 5 burst sequence. For physiological $Ca^{2+}$ dependence studies, each cell was normalized to its 100 AP 10 Hz behavior. All statistical tests were done using OriginPro (OriginLab), all statistical tests were two-sided. One sample *t*-test, and paired t-test were used for internal comparisons for changes

within an individual cell depending upon normalization; unless otherwise stated data met the test criteria and was not transformed. Kolmogorov-smirnov tests were used to compare time constants between cells as has been previously established (*Armbruster and Ryan, 2011*). Dynamin 1 rescue expressed in dynamin 1/3 DKO had 2 outliers which were >3 standard deviations away from the mean at 28 s and 36 s. These cells were treated as outliers and are not included.

## Acknowledgements

The authors would like to thank Jeremy Dittman and members of the Ryan laboratory for valuable discussions. We thank Yogesh Gera for excellent technical assistance. vG-pH was a gift from Rob Edwards and Susan Voglamaier (UCSF). This work was supported in part by NIH grants to TAR (NS036942) and PDC (R37NS036251 and DA018343) and the David Rockefeller Graduate School at the Rockefeller University.

## Additional information

### Funding

| Funder | Grant reference number | Author |
| --- | --- | --- |
| National Institute for Neurological Disorders and Stroke | NS036942 | Timothy A Ryan |
| Howard Hughes Medical Institute | | Pietro De Camilli |
| National Institute for Neurological Disorders and Stroke | NS036251 | Pietro De Camilli |
| National Institutes of Health | DA018343 | Pietro De Camilli |

The funders had no role in study design, data collection and interpretation, or the decision to submit the work for publication.

### Author contributions

MA, Conception and design, Acquisition of data, Analysis and interpretation of data, Drafting or revising the article; MM, PDC, Conception and design, Drafting or revising the article, Contributed unpublished essential data or reagents; SMF, Conception and design, Contributed unpublished essential data or reagents; TAR, Conception and design, Analysis and interpretation of data, Drafting or revising the article

### Ethics

Animal experimentation: This study was performed in strict accordance with the recommendations in the Guide for the Care and Use of Laboratory Animals of the National Institutes of Health. All of the animals were handled according to approved institutional animal care and use committee (IACUC) protocols (#0601-450A) of Weill Cornell Medical College. The protocol was approved by the Committee on the Ethics of Animal Experiments at the Weill Cornell Medical College. Every effort was made to minimize suffering.

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
