## [Decision Letter]

Thank you for sending your work entitled “Dynamin phosphorylation controls optmization of endocytosis for brief action potential bursts” for consideration at *eLife*. Your article has been favorably evaluated by a Senior editor and 3 reviewers, one of whom, Graeme Davis, is a member of our Board of Reviewing Editors.

The Reviewing editor and the other reviewers discussed their comments before we reached this decision, and the Reviewing editor has assembled the following comments to help you prepare a revised submission.

The paper by Armbruster et al. focuses on the kinetics of synaptic vesicle endocytosis at the nerve terminal. This is an important biological problem and the present paper reports on a new cellular/molecular mechanism by which synaptic vesicle endocytosis is accelerated in an activity-dependent manner. The approach to study endocytosis by following VGluT1-pHluorin fluorescence has been validated and is powerful. The study should appeal to the broad readership of *eLife*.

Major comments:

1) The authors employ a powerful strategy of transgene expression in a dynamin knockout background. As a control, it is important to determine whether or not the KO of dynamin 1 itself affects the observed acceleration of endocytosis rates (e.g., please add data on WT and dynamin 1 KO to Figure 4).

2) The authors probe the consequences of altered stimulus number. The effect of stimulation frequency should be addressed. For example, does one see acceleration for 25APs at 40Hz vs. 100 APs at 40Hz?

3) There has been debate among the reviewers regarding whether it is necessary to show an effect on short-term synaptic plasticity. As highlighted by the authors in the Introduction, an implication of altered endocytic rate is that it will impact the dynamics of short-term synaptic release properties. However, the reviewers acknowledge that this study focuses on endocytic mechanisms, and that future studies will be necessary to explore the impact on short-term synaptic plasticity. In particular, determining the precise contribution of changing endocytic rate to short-term release dynamics will not be trivial, in part because it will be necessary to dissociate the effects of altered endocytic rate from other influences on release dynamics including calcium dynamics and vesicle pool dynamics. As such, this is best left to a future study.

However, it does seem appropriate to include more explicit control data regarding basic properties of synaptic vesicle release that could be understood from existing pHluorin experiments in the existing experimental system. This should be done to control for effects on gross neurotransmitter release and/or pool sizes in the wild type and transgenic rescue experiments. Please show this information for neurons expressing WT dynamin 1, phosphomimetic dynamin 1, and phosphorylation-deficient dynamin 1.

---

## [Author Response]

*1) The authors employ a powerful strategy of transgene expression in a dynamin knockout background. As a control, it is important to determine whether or not the KO of dynamin 1 itself affects the observed acceleration of endocytosis rates (e.g., please add data on WT and dynamin 1 KO to*
Figure 4*)*.

Our reasoning for using the dynamin 1/3 KO as the background in which we would reintroduce mutant dynamins is based on the following:

A) The single dynamin 1 KO had a remarkably subtle phenotype (Ferguson et al. Science 2007; Raimondi et al Neuron 2011) owing to the fact that there is sufficient dynamin 3 to allow normal endocytosis. Only upon removal of forms does one see a significant impact on endocytosis.

B) While less well characterized, dynamin 3 also contains a similar phospho-box motif as dynamin 1 (Larsen et al. Mol. Cell Proteomics 2004). As a result we felt that any result that would be obtained in the dynamin-1 KO would be difficult to interpret until the phosphorylation cycle of dynamin 3 has been better characterized. We have attempted to clarify this in the text.

Regarding including control data in Figure 4: we have made many versions of this figure trying to decide what to include. While we agree with the reviewers’ suggestion that is would enhance the comparison, visually it makes it difficult with all 4 overlapping traces. Since the control trace is already included in Figure 4, we have left it unchanged.

*2) The authors probe the consequences of altered stimulus number. The effect of stimulation frequency should be addressed. For example, does one see acceleration for 25APs at 40Hz vs. 100 APs at 40Hz*?

We have now included a dataset for 30 Hz stimulation. As expected this condition shifts the minimum endocytosis time to smaller stimuli and makes it difficult to resolve, similar to what happened at lower temperatures. This endocytosis curve (10,15,25,50,100AP) at 30Hz is included in Figure 2—figure supplement 2. The acceleration is clearly present when compared to the 1AP time constant; however, the minimum is below 15AP and cannot be clearly resolved as expected for conditions that lead to greater accumulation of intracellular calcium.

*3) There has been debate among the reviewers regarding whether it is necessary to show an effect on short-term synaptic plasticity. As highlighted by the authors in the Introduction, an implication of altered endocytic rate is that it will impact the dynamics of short-term synaptic release properties. However, the reviewers acknowledge that this study focuses on endocytic mechanisms, and that future studies will be necessary to explore the impact on short-term synaptic plasticity. In particular, determining the precise contribution of changing endocytic rate to short-term release dynamics will not be trivial, in part because it will be necessary to dissociate the effects of altered endocytic rate from other influences on release dynamics including calcium dynamics and vesicle pool dynamics. As such, this is best left to a future study*.

*However, it does seem appropriate to include more explicit control data regarding basic properties of synaptic vesicle release that could be understood from existing pHluorin experiments in the existing experimental system. This should be done to control for effects on gross neurotransmitter release and/or pool sizes in the wild type and transgenic rescue experiments. Please show this information for neurons expressing WT dynamin 1, phosphomimetic dynamin 1, and phosphorylation-deficient dynamin 1*.

We have included controls on the recycling pool size and the vesicle release rate for the various dynamin 1-rescue conditions in Figure 4—figure supplement 1.